# Examining the liver–pancreas crosstalk reveals a role for the molybdenum cofactor in β-cell regeneration

Christos Karampelias[1,2], Bianca Băloiu[1], Birgit Rathkolb[3,4,5], Patricia da Silva-Buttkus[3], Etty Bachar-Wikström[1], Susan Marschall[3], Helmut Fuchs[3], Valerie Gailus-Durner[3], Lianhe Chu[1], Martin Hrabě de Angelis[3,5,6], Olov Andersson[1,7]

Regeneration of insulin-producing β-cells is an alternative avenue to manage diabetes, and it is crucial to unravel this process in vivo during physiological responses to the lack of β-cells. Here, we aimed to characterize how hepatocytes can contribute to β-cell regeneration, either directly or indirectly via secreted proteins or metabolites, in a zebrafish model of β-cell loss. Using lineage tracing, we show that hepatocytes do not directly convert into β-cells even under extreme β-cell ablation conditions. A transcriptomic analysis of isolated hepatocytes after β-cell ablation displayed altered lipid- and glucose-related processes. Based on the transcriptomics, we performed a genetic screen that uncovers a potential role of the molybdenum cofactor (Moco) biosynthetic pathway in β-cell regeneration and glucose metabolism in zebrafish. Consistently, molybdenum cofactor synthesis 2 (*Mocs2*) haploinsufficiency in mice indicated dysregulated glucose metabolism and liver function. Together, our study sheds light on the liver–pancreas crosstalk and suggests that the molybdenum cofactor biosynthesis pathway should be further studied in relation to glucose metabolism and diabetes.

## Introduction

Diabetes is a multisystemic disease projected to affect up to 640 million people worldwide by 2040 (Sun et al, 2022). Insulin-producing pancreatic β-cells have a central role in diabetes pathophysiology. Patients with type 1 diabetes suffer from autoimmune-mediated depletion of their β-cells, whereas patients with type 2 diabetes manifest with reduced functional β-cell mass in later stages of the disease (Bakhti et al, 2019; Atkinson & Mirmira, 2023). Recent advances in diabetes research have shown intricate crosstalk between multiple tissues and β-cells that can regulate

β-cell mass and function (Shirakawa et al, 2017; Wang et al, 2019; Chiou et al, 2021; Kahraman et al, 2022; Atkinson & Mirmira, 2023). Therefore, understanding how β-cell mass and function can be regulated by signals emanating from various tissues can increase our understanding of diabetes development and progression, as well as generating novel therapies for the disease.

Stimulating β-cell regeneration holds interest as an approach to develop new therapeutics for diabetes patients. Important advances in understanding endogenous β-cell regeneration have been made possible using animal models. In these models, typically, there is an induced loss of β-cell mass that has led to studies into the mechanism of β-cell regeneration. A near-complete ablation of β-cells led to reprogramming of other pancreatic cell types including α-, δ-, γ-, or acinar cells to β-cells (Thorel et al, 2010; Chera et al, 2014; Ye et al, 2015; Lu et al, 2016; Druelle et al, 2017; Lee et al, 2018; Furuyama et al, 2019; Perez-Frances et al, 2021; Carril Pardo et al, 2022; Singh et al, 2022). Proliferation of remaining β-cells also contributed to β-cell regeneration in less profound injury models (reviewed in Basile et al [2022]). A controversial third regenerative pathway relies on pancreatic tissue-resident stem cell differentiation to endocrine fate, findings that are heavily debated for their applicability and can differ between injury models and model organisms (Xu et al, 2008; Solar et al, 2009; Kopp et al, 2011; Van de Casteele et al, 2013; Xiao et al, 2013; Sancho et al, 2014; Delaspre et al, 2015; Ghaye et al, 2015; Wang et al, 2020; Gribben et al, 2021; Zhao et al, 2021; Karampelias et al, 2022; Magenheim et al, 2023). Importantly, few of these studies have investigated how other organs and cell types outside the pancreas regulate regeneration of the β-cell mass.

Limited examples of signals from other tissues that contribute to β-cell regeneration exist (Shirakawa et al, 2017), e.g. adiponectin, an adipocyte-derived hormone, stimulated β-cell regeneration in mice (Ye et al, 2014). The liver is another central hub integrating molecular signals from the pancreatic tissue (Titchenell et al, 2017). The crosstalk is evident as circulating factors from the liver can

---

[1]Department of Cell and Molecular Biology, Karolinska Institutet, Stockholm, Sweden   [2]Institute of Diabetes and Regeneration Research, Helmholtz Munich, Neuherberg, Germany   [3]Institute of Experimental Genetics, German Mouse Clinic, Helmholtz Zentrum München, Neuherberg, Germany   [4]Institute of Molecular Animal Breeding and Biotechnology, Gene Center, Ludwig-Maximilians-Universität München, Munich, Germany   [5]German Center for Diabetes Research (DZD), Neuherberg, Germany   [6]Chair of Experimental Genetics, TUM School of Life Sciences, Technische Universität München, Freising, Germany   [7]Department of Medical Cell Biology, Uppsala University, Uppsala, Sweden

Correspondence: christos.karampelias@helmholtz-munich.de; olov.andersson@ki.se, olov.andersson@mcb.uu.se
Etty Bachar-Wikström's present address is Dermatology and Venereology Division, Department of Medicine, Karolinska Institutet, Stockholm, Sweden

---

potentially regulate both α- and β-cell mass. First, a mouse model of hepatocyte-specific insulin receptor knockout showed an increase in β-cells (Michael et al, 2000; El Ouaamari et al, 2013). Using this mouse model, Serpinb1 has been identified as one of the circulating factors that drives β-cell hyperplasia and correlating with insulin sensitivity (El Ouaamari et al, 2016; Glicksman et al, 2017). Second, additional in vivo models pointed to Igfbp1 as a secreted protein from the liver that promoted α-cell-to-β-cell transdifferentiation, kisspeptin as a liver-secreted neuropeptide that regulated insulin secretion from β-cells, and FGF21 as a liver-secreted cytokine that mediates β-cell regeneration in a mouse model treated with a glucagon receptor–blocking antibody (Song et al, 2014; Lu et al, 2016; El Ouaamari et al, 2019; Cui et al, 2023). Third, certain amino acids secreted from the liver promoted α-cell hyperplasia in mouse islets (Solloway et al, 2015). Fourth, Pdx1 overexpression in mouse hepatocytes triggered a direct hepatocyte-to-β-cell conversion (Ferber et al, 2000; Meivar-Levy & Ferber, 2019). Taken together, these results suggest a promising but understudied role of the liver in stimulating β-cell regeneration under diabetic conditions.

In this study, we aimed to explore how the liver and more specifically the hepatocytes contribute to β-cell regeneration in zebrafish. Zebrafish can regenerate most of its tissues after injury, and that is also true for their β-cell population (Goode et al, 2022; Mi et al, 2024). Here, we used the nitroreductase (NTR)–metronidazole (MTZ) zebrafish model to induce a near-complete β-cell ablation and study how the hepatocytes could be involved in the β-cell regeneration response (Curado et al, 2007; Pisharath et al, 2007). We showed that there is no spontaneous hepatocyte-to-β-cell conversion in the regenerating zebrafish larvae. Moreover, we characterized the transcriptome of zebrafish hepatocytes after β-cell ablation with the aim of identifying immediate secreted signals or enzymes generating metabolites that can stimulate β-cell regeneration. We found that the overexpression of the short isoform of the molybdenum cofactor synthesis 2 (Mocs2) enzyme (coding for the catalytic subunit of the molybdopterin synthase complex) in the hepatocytes could reduce glucose levels and stimulate a small but significant increase in β-cell regeneration. Mocs2 participates in the synthesis of the molybdenum cofactor (Moco) by catalyzing the synthesis of molybdopterin (Mayr et al, 2021), and we found that treatment with a molybdenum source, sodium molybdate, led to an increase in β-cell regeneration, albeit the phenotype was variable. In translating these findings to mice, we found that Mocs2 haploinsufficient male mice manifested with a trend of impaired glucose tolerance. Together, our work has characterized important aspects of how the liver can affect β-cell regeneration and glucose homeostasis, and highlighted a role for Moco in this context.

## Results

### Characterization of the hepatocytes' contribution to the spontaneous β-cell regeneration in zebrafish

The aim of this study is to characterize how the hepatocytes, which comprise the largest cell population of the liver, can contribute to

pancreatic β-cell regeneration. We modeled β-cell loss using the NTR-MTZ ablation system in zebrafish. Briefly, in this transgenic zebrafish model the β-cells express the enzyme NTR. When the prodrug MTZ is added to the zebrafish water, NTR converts MTZ to a toxic by-product and specifically ablates the β-cells. Using this β-cell ablation system, we explored the possibility that hepatocytes could be reprogrammed to β-cells, as documented previously, but without the forced expression of the Pdx1 transcription factor (Meivar-Levy & Ferber, 2019). To this end, we used the Tg(fabp10a: Cre);Tg(ubi:Switch) zebrafish (Fig 1A). In this lineage-tracing system, the fabp10a-expressing hepatocytes are permanently labeled by mCherry after recombination with the hepatocyte-specific Cre driver. We ablated the β-cells of zebrafish larvae with the addition of MTZ from 3 to 4 days post-fertilization (dpf) and let the larvae regenerate for 2 d, a procedure that we from now on refer to as a β-cell regeneration assay. We did not observe any hepatocyte-derived β-cells in the pancreas after 2 d of regeneration (Fig 1B and B'). Moreover, there were no insulin-producing cells present in the liver with or without β-cell ablation (Fig 1C–D'). These results demonstrate that hepatocytes are not spontaneously reprogrammed to β-cells in the highly regenerative developing zebrafish.

Next, we aimed to damage the hepatocytes and assess whether that leads to a reduction in β-cells in the pancreas of zebrafish larvae. To this end, we treated the zebrafish larvae with a high concentration of acetaminophen, a drug that has previously been shown to induce hepatocyte damage in zebrafish (North et al, 2010). We confirmed that 24 h of acetaminophen treatment reduced liver size in Tg(fabp10a:GFP) zebrafish larvae. Moreover, we observed that β-cell ablation reduced liver size to comparable levels as acetaminophen treatment (Fig 1E–I). This effect was independent of MTZ treatment as larvae without the Tg(ins:flag-NTR) transgene but treated with MTZ had normal liver size. Conversely, we explored whether hepatocyte damage/acetaminophen treatment impaired β-cell development; acetaminophen treatment for 24 h did not affect β-cell development of 5 dpf zebrafish (Fig 1J–L). We corroborated these results with a genetic model of hepatocyte ablation based on the same NTR-MTZ ablation system. Transgenic Tg(fabp10a:CFP-NTR) zebrafish larvae were treated with MTZ to ablate almost all hepatocytes, as shown before (Choi et al, 2014). Similar to acetaminophen treatment, this did not lead to any changes in β-cell development (Fig S1A–C). We also attempted to simultaneously ablate hepatocytes and β-cells using either the acetaminophen treatment in the Tg(ins:flag-NTR) background or the MTZ-NTR system in the double transgenic Tg(fabp10a:CFP-NTR);Tg(ins:flag-NTR) zebrafish. However, the double ablation system was not tolerated and embryonic lethal, prohibiting us from assessing the double ablation phenotype. Overall, our data show that no hepatocyte-to-β-cell reprogramming is observed in zebrafish larvae and that liver damage does not affect β-cell development.

### Transcriptomic changes in hepatocytes after β-cell ablation

Secreted proteins from the liver have been shown to contribute to compensatory β-cell expansion in zebrafish and mice. Therefore, we decided to explore the possibility that additional hepatocyte-derived secreted signals can stimulate β-cell expansion. To this end, we characterized the transcriptome of 2-mo-old zebrafish hepatocytes after β-cell ablation. In this experiment, we ablated

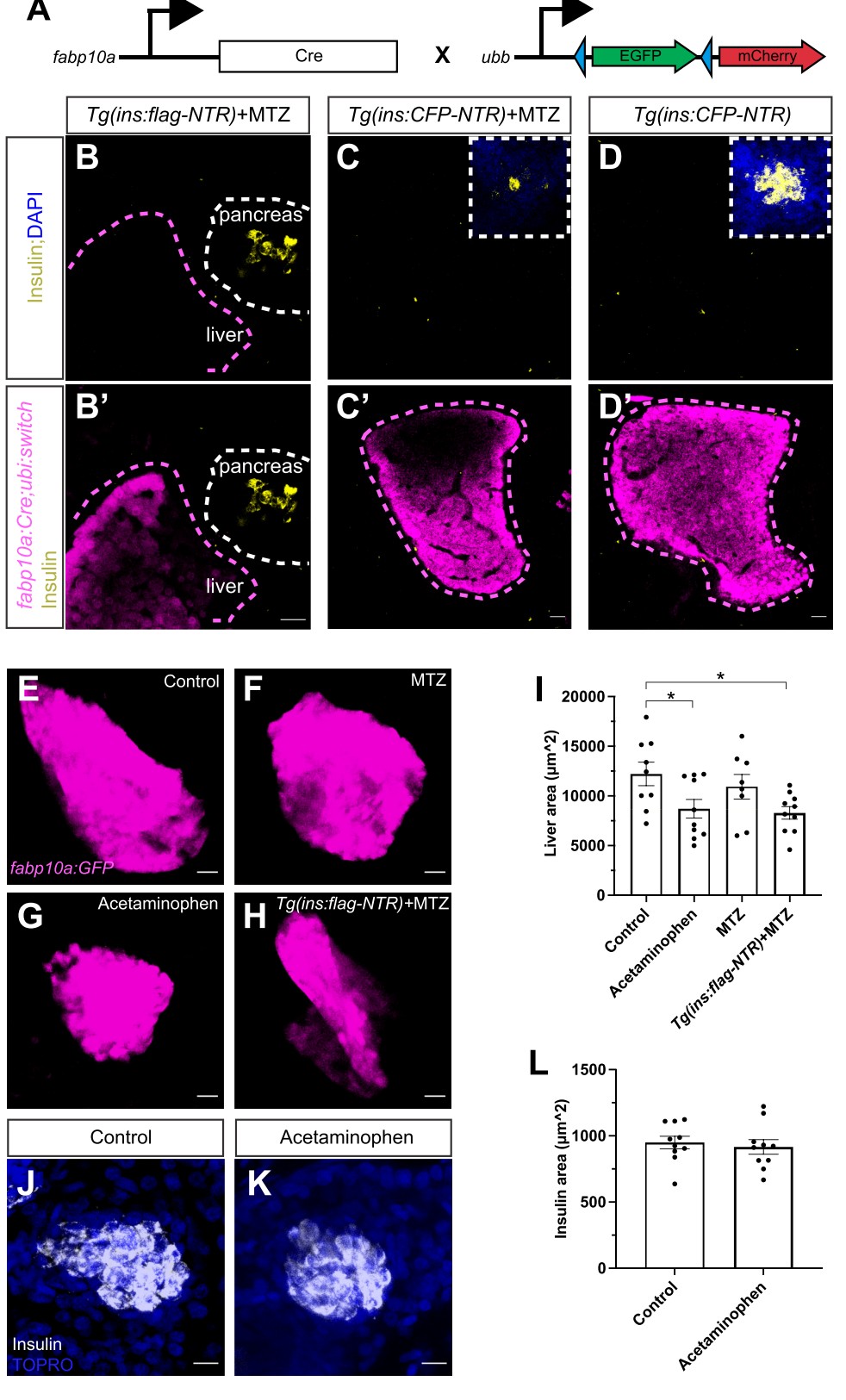

**Figure 1. Hepatocytes' contribution to the spontaneous β-cell regeneration in zebrafish.**
**(A)** Schema showing the lineage-tracing approach to characterize the hepatocyte-to-β-cell reprogramming using the *Tg(fabp10a:Cre);Tg(ubi:Switch)* zebrafish. Blue arrowheads indicate the loxP sites. **(B, B′)** Single-plane confocal images of the *Tg(fabp10a:Cre);Tg(ins:flag-NTR);Tg(ubi:switch)* pancreas (B) and liver (B′) of 6 dpf zebrafish larvae after 2 d of β-cell regeneration immunostained against insulin. The white dashed line outlines the pancreas, and the magenta dashed line outlines the border of the liver. Scale bar, 20 μm. *n* ≥ 10 larvae examined from two independent experiments. **(C, C′, D, D′)** Single-plane confocal images of liver and primary islets of *Tg(fabp10a:Cre);Tg(ins:CFP-NTR);Tg(ubi: switch)* 6 dpf larvae, with (C, C′) or without (D, D′) MTZ treatment, that is, β-cell ablation. Larvae in (C, C′) were left to regenerate their β-cells for 2 d. **(C, C′, D, D′)** White dashed line outlines the insets of the primary islet of the pancreas (C, D), and the magenta dashed line outlines the border of the liver (C′, D′). Scale bar, 20 μm. *n* ≥ 10 larvae examined from two independent replicates. **(E, F, G, H, I)** Maximum projections of livers from control (E), MTZ-treated (F), acetaminophen-treated (G), and *Tg(ins:flag-NTR)*+MTZ-treated (H) *Tg(fabp10a:GFP)* 4 dpf zebrafish larvae. Chemical treatments were carried out at 3–4 dpf. **(I)** Quantification showed a significant decrease in the hepatocyte area after hepatocyte damage or β-cell ablation (I). Scale bar, 20 μm. *n* = 8–10. Data are presented as the mean values ± SEM. One-way ANOVA was used to estimate statistical significance followed by a Holm–Šidák multiple comparison test. *P = 0.0325 (control versus acetaminophen); *P = 0.0237 (control versus *Tg(ins:flag-NTR)*+MTZ). **(J, K, L)** Representative maximum projections of pancreatic islets in control (J) and acetaminophen-treated (3–4 dpf) (K) 5 dpf zebrafish larvae immunostained against insulin. Nuclei were counterstained with TO-PRO-3. **(L)** Quantification of the insulin area (L). Scale bar, 10 μm. *n* = 10.

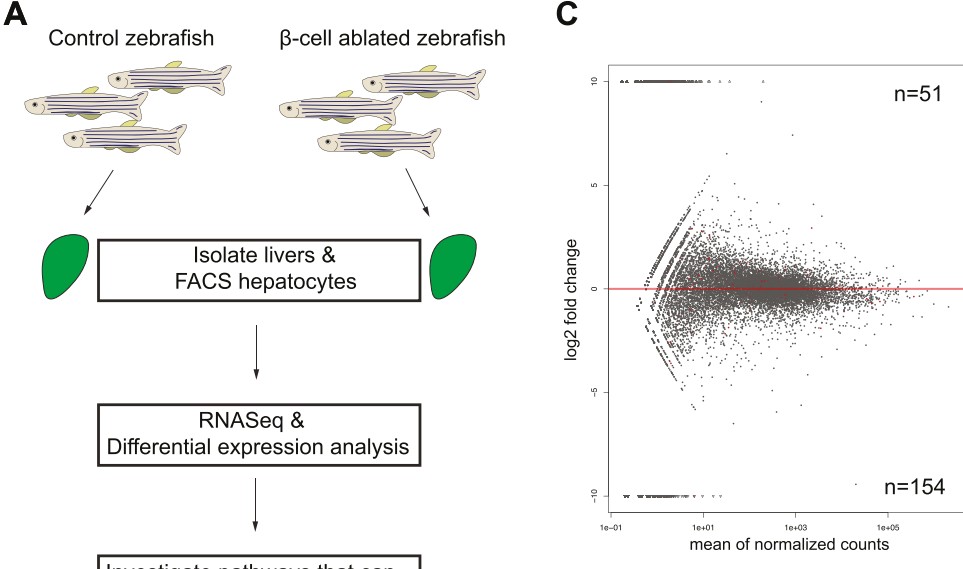

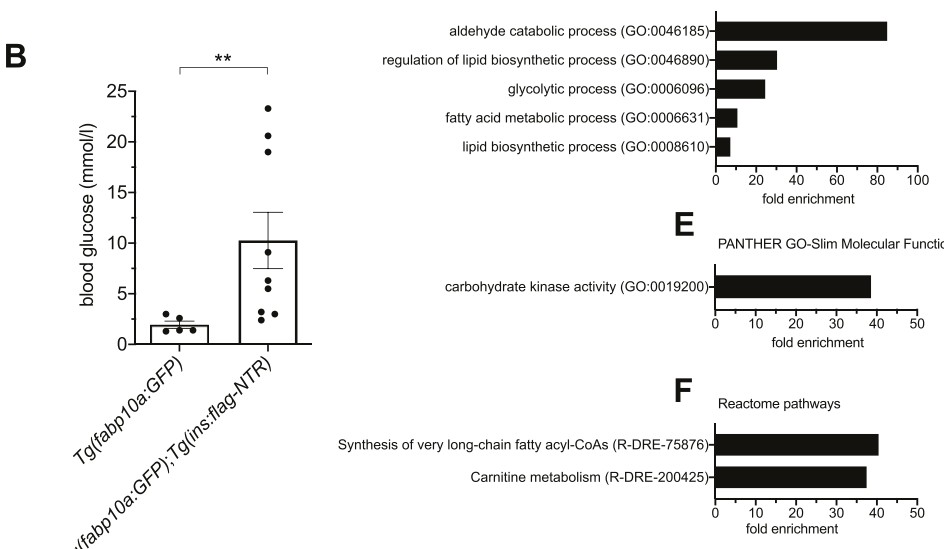

**Figure 2. Transcriptomic changes in hepatocytes after β-cell ablation.**
**(A)** Schema showing the experimental design to identify transcriptional changes in hepatocytes after β-cell ablation in 2-mo-old zebrafish. **(B)** Glucose measurements of the 2-mo-old zebrafish used for hepatocyte isolation. Measurements were made before liver dissection. $n$ = 5–9. A Mann–Whitney test was used to assess significance. **$P$ = 0.0045. **(C)** Log$_2$ fold change plot showing the up-regulated and down-regulated genes in hepatocytes after β-cell ablation. The significantly differentially expressed genes ($P$ adj < 0.05) are highlighted as red dots. **(C, D, E, F)** Statistical overrepresentation analysis of the down-regulated genes from (C) using the PantherDB tool. **(D, E, F)** Fold enrichment of the significantly enriched processes (D), molecular function (E), and Reactome pathways (F) is shown.

β-cells in our zebrafish model followed by 1 d of regeneration, and then euthanized the zebrafish and dissected out the livers. We used the *Tg(fabp10a:GFP)* as the control samples and *Tg(fabp10a:GFP); Tg(ins:flag-NTR)* zebrafish as the β-cell–ablated samples. Then, we FACS-sorted the GFP⁺ hepatocytes, extracted RNA, and performed bulk RNA-Seq (Fig 2A). Furthermore, we confirmed that the β-cell ablation was successful by measuring the blood glucose levels at the time of euthanizing the zebrafish (Fig 2B).

We performed differential expression analysis to characterize the transcriptomic changes in the hepatocytes after β-cell ablation. Overall, we observed a greater number of down-regulated genes compared with up-regulated genes (Fig 2C). Key genes that have

previously been shown to be regulated by insulin were down-regulated in hepatocytes after β-cell ablation (Supplemental Data 1). For example, glucokinase (*gck*) was the most significantly down-regulated gene in our dataset. Moreover, we observed a non-significant up-regulation of both *serpinb1* and *igfbp1a*, two secreted peptides that were previously been implicated in β-cell regeneration (Fig S2A). Subsequently, we performed a gene ontology enrichment analysis to reveal the pathways that were significantly affected in hepatocytes after β-cell ablation. In general, cellular processes and metabolic processes were two of the most affected biological processes in both up- and down-regulated genes (Fig S2B and C). We next performed a statistical overrepresentation analysis

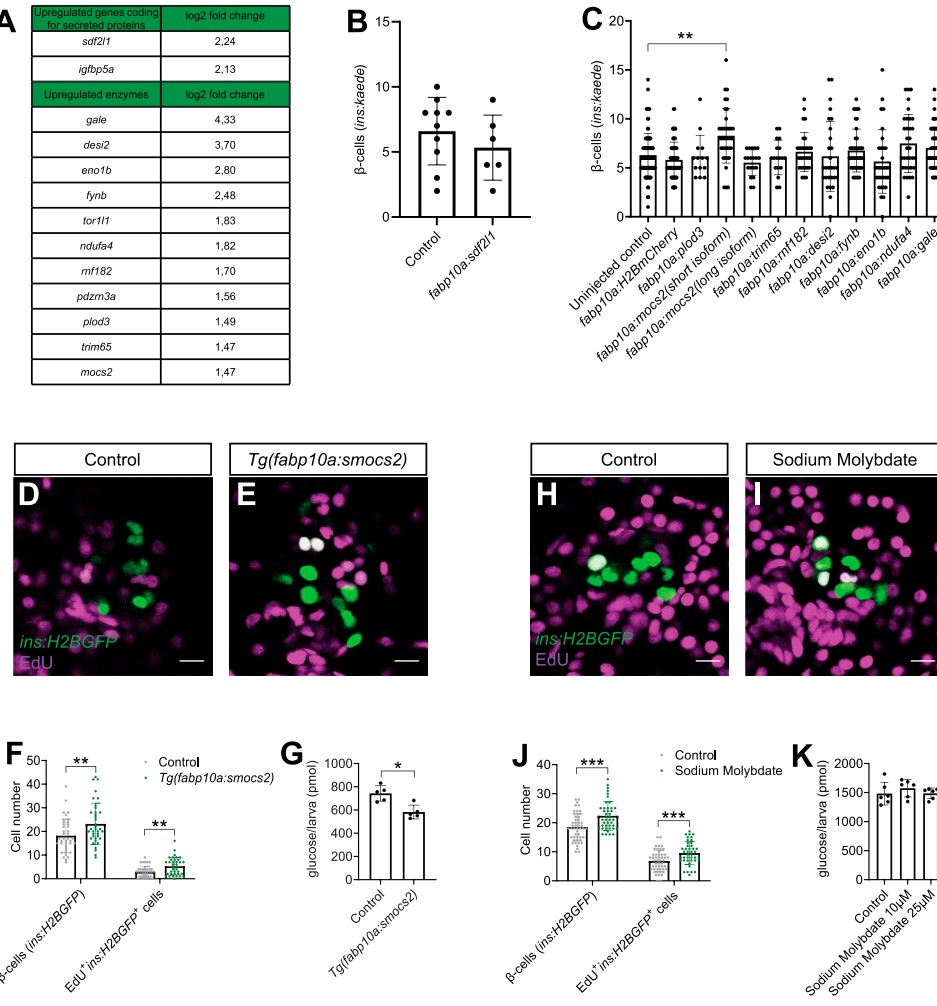

**Figure 3. Genetic screen reveals a role for the molybdenum cofactor biosynthetic pathway in β-cell regeneration.**
**(A)** Table showing the log$_2$ fold changes of the significantly up-regulated genes encoding for secreted proteins and enzymes in hepatocytes after β-cell ablation. **(B)** Quantification of regenerated β-cells in 6 dpf *Tg(ins:kaede);Tg(ins:CFP-NTR)* larvae overexpressing *sdf2l1* in hepatocytes. *n* = 6–10. Data are presented as mean values ± SEM. **(A, C)** Quantification of β-cells in control and zebrafish larvae injected at the one-cell stage with vectors driving the expression of the enzymes identified in hepatocytes (A) under the control of the *fabp10a* promoter (together with mRNA encoding the transposase enzyme to induce genomic integration). After injections, β-cells were ablated from 3 to 4 dpf and the β-cells were counted manually after 2 d of regeneration in the *Tg(ins:kaede); Tg(ins:CFP-NTR)* zebrafish larvae at 6 dpf. *n* = 14–104. Data for the control experiments were pooled from four independent experiments. If there was a positive hit in the first experiment, the experiments were repeated and data pooled in this graph. A Kruskal–Wallis test followed by Dunn's multiple comparison test was used to assess statistical significance. **P = 0.0038. Data are presented as mean values ± SEM. **(D, E, F)** Single-plane confocal images of pancreatic islets of control (D) and *Tg(fabp10a:smocs2)* (E) *Tg(ins:flag-NTR);Tg(ins: H2BGFP)* 6 dpf zebrafish larvae after 2 d of β-cell regeneration, during which EdU incubation occurred. **(F)** Quantification of the number of *ins:H2BGFP*+ and *ins: H2BGFP*+EdU+ cells (F). Scale bar, 10 μm. *n* = 37–40. A Mann–Whitney test was used to assess significance. **P = 0.0079 and 0.0037, respectively. Data are presented as mean values ± SEM and are pooled from three

independent experiments. **(G)** Glucose levels in *Tg(ins:flag-NTR);Tg(fabp10a:smocs2)* 6 dpf larvae. Four larvae were pooled for each replicate. *n* = 5. A Mann–Whitney test was used to assess significance. *P = 0.0159. Data are presented as mean values ± SEM. **(H, I, J)** Single-plane confocal images of pancreatic islets of control (H) and sodium molybdate–treated (I) *Tg(ins:flag-NTR);Tg(ins:H2BGFP)* 6 dpf zebrafish larvae after 2 d of β-cell regeneration, during which EdU incubation occurred. **(J)** Quantification of the number of *ins:H2BGFP*+ and *ins:H2BGFP*+EdU+ cells (J). Scale bar, 10 μm. *n* = 42–49. A Mann–Whitney test was used to assess significance for β-cell numbers, and an unpaired *t* test was used to assess significance for *ins:H2BGFP*+EdU+. ***P = 0.0005 and 0.0008, respectively. Data are presented as mean values ± SEM and are pooled from three independent experiments. **(K)** Glucose levels of *Tg(ins:CFP-NTR)* 6 dpf larvae treated with 10 or 25 μM sodium molybdate. Four larvae were pooled for each replicate. *n* = 6. Data are presented as mean values ± SEM.

and found no significantly enriched gene categories for the up-regulated genes. On the contrary, glycolytic and lipid-related biological processes were significantly enriched in the down-regulated genes (Fig 2D). In addition, down-regulated genes were enriched in carbohydrate kinase activity and in the synthesis of very-long-chain fatty acyl-CoAs and carnitine metabolism Reactome pathways (Fig 2E and F). Overall, our transcriptomic characterization of hepatocytes after β-cell ablation in 2-mo-old zebrafish showed a down-regulation of glycolytic and lipid-related metabolic processes.

**An in vivo genetic screen to identify hepatocyte genes that can enhance β-cell regeneration**

Next, we focused our analysis on the significantly up-regulated genes in the RNA-Seq dataset. We aimed to determine whether the up-regulated genes coding for secreted proteins or enzymes could accelerate β-cell regeneration when overexpressed in hepatocytes to supraphysiological levels. Our dataset contained two genes coding for secreted proteins and 11 genes coding for enzymes that were significantly up-regulated (*P* adj < 0.05) (Fig 3A). We cloned these genes under the control of the liver-specific *fabp10a* promoter and injected them in the *Tg(ins:kaede);Tg(ins:CFP-NTR)* zebrafish model to perform our standard β-cell regeneration assay. Of note, we did not perform the regeneration assay with the *igfbp5a* gene as we tested it in our previously published work (Lu et al, 2016). The mosaic overexpression of the single up-regulated gene coding for a secreted protein, *sdf2l1*, did not increase β-cell regeneration (Fig 3B). Subsequently, we overexpressed nine of the 11 significantly up-regulated enzymes (as we did not manage to clone two of the enzymes, namely, *pdzrn3a* and *tor1l1*; see the Materials and Methods section) and discovered a modest, yet significant, increase

in β-cell regeneration upon the overexpression of the short isoform of the *mocs2* gene. On the contrary, overexpressing the long isoform of the *mocs2* enzyme did not result in a similar phenotype, suggesting different functions of the two protein isoforms (Fig 3C). Overall, our genetic screen pointed to the short isoform of *mocs2* as a gene candidate that could be implicated in β-cell regeneration via its expression in the liver.

## Phenotypic characterization of *mocs2* overexpression in zebrafish

After our genetic screen, we generated a stable transgenic line overexpressing the short isoform of *mocs2* in hepatocytes, named from here on *Tg(fabp10a:smocs2)*. We observed that the *Tg(fabp10a:smocs2)* larvae increased β-cell regeneration similar to the mosaic overexpression (Fig 3D–F). The extent of β-cell regeneration was variable between experimental replicates with some showing a profound increase in β-cell regeneration, whereas other experiments showed a less pronounced phenotype that did not reach statistical significance. To study the cellular mechanism behind the observed β-cell regeneration, we incubated control and *Tg(fabp10:smocs2)* larvae with EdU during the regenerative period to assess β-cell proliferation. We observed a cumulative significant increase of EdU incorporation in regenerating β-cells (control: 16.6% ± 1.97 SEM versus *Tg(fabp10a:smocs2)*: 22% ± 2.18 SEM proliferating β-cells), yet the phenotype was variable between replicates similar to the β-cell regeneration observation (Fig 3D–F). This suggests that there are other biological factors that need to coordinate with the *mocs2* overexpression to reach the full potential of the phenotype. In addition, we assessed the glucose levels in the *Tg(fabp10a:smocs2)* larvae. We noticed a decrease of glucose levels in larvae on the *Tg(fabp10:smocs2)* background compared with control larvae after β-cell ablation, a phenotype that was more consistent than the increase in β-cell regeneration (Fig 3G).

Next, we aimed to stabilize and clarify the effect of Moco biosynthesis in β-cell regeneration using a non-genetic approach. To this extent, we treated zebrafish larvae with sodium molybdate, which has previously been used as a source of molybdenum in *E. coli* cultures and can rescue defects in the Moco biosynthetic pathway (Warnhoff & Ruvkun, 2019), for 2 d after β-cell ablation. Sodium molybdate increased β-cell regeneration through increased β-cell proliferation (control: 35.6% ± 2.07 SEM versus sodium molybdate: 42.2% ± 2.22 SEM proliferating β-cells), similar to the overexpression of the short isoform of *mocs2*, yet the phenotype also appeared to be variable resembling the genetic model (Fig 3H–J). To clarify the variability between experiments, we present all independent biological experiments showing β-cell regeneration and proliferation in Fig S3A–F. Similar results were obtained when using a β-cell–specific FUCCI reporter zebrafish (*ins:venus-geminin*) (Sakaue-Sawano et al, 2008; Lu et al, 2016) that reports cell cycle progression (Fig S4A–C). EdU incorporation in β-cells was unchanged by sodium molybdate treatment during development, that is, without β-cell ablation (Fig S4D–F), and ablation of β-cells did not affect α- or δ-cell proliferation (Fig S4G–J). Contrary to the genetic overexpression of *mocs2*, none of the experiments with sodium molybdate treatments showed decreased glucose levels, highlighting that there are both similarities and differences between the genetic and chemical approaches (Fig 3K). Mechanistically, the

gene expression of the core molybdenum biosynthesis pathway was unaltered in both the *Tg(fabp10a:smocs2)*- and the sodium molybdate–treated zebrafish (Fig S5A–J and M–V). No gene expression changes were observed in key glycolytic/gluconeogenic enzymes (Fig S5K, L, and W–X). Yet, the *Tg(fabp10a:smocs2)* zebrafish showed a strong tendency for down-regulation of the *suox* and *xdh* Moco-using enzymes at the end of the regeneration assay (Fig S5I and J), suggesting a level of transcriptional compensation for the molybdenum cofactor biosynthetic pathway and/or potentially explaining the difference in the glucose-lowering effect between the genetic model and chemical treatment. Overall, our combinatorial chemical/genetic approach to perturb the Moco biosynthetic pathway showed a role for this pathway in β-cell biology and glucose homeostasis, but additional factors are needed to recapitulate a more robust phenotype.

## Gene expression of the core genes of the Moco biosynthetic pathway across species

To better understand the Moco pathway, we assessed the expression levels of the genes involved in Moco biosynthesis, as well as the enzymes that use the molybdenum cofactor for their function across species. In zebrafish, we identified orthologues of all the mammalian genes involved in Moco biosynthesis, namely, *mocs1*, *mocs2*, *mocs3*, *gphna*, and *gphnb*, together with homologues of enzymes requiring molybdenum for their function including *suox*, *xdh*, *aox5*, and *aox6*. All enzymes involved in Moco biosynthesis and use were expressed in our RNA-Seq dataset of isolated hepatocytes except for *gphnb* (the paralogue of *ghpna*) and *aox5* (Fig S6A). Comparing how their expression was affected by β-cell ablation, we observed that only *mocs2* was significantly up-regulated with the rest of the enzymes' expression not significantly affected (Fig S6B). We then assessed the expression levels of the same enzymes in our RNA-Seq of isolated islets from zebrafish larvae with or without β-cell ablation, data that we previously published (Karampelias et al, 2021). We observed a similar expression level for all enzymes of the pathway except for *gphnb*, which had expression below the level of detection, similar to its absence of expression in hepatocytes (Fig S6C). Furthermore, β-cell ablation did not significantly up-regulate the expression of any genes in the pathway, with the most noticeable change being a down-regulation of the *suox* enzyme (Fig S6D). Expression levels of *ins* were used as a reference for both datasets (Fig S6A–D).

We used publicly available single-cell RNA-Seq (scRNA-Seq) datasets and atlases to assess the expression of the genes involved in the molybdenum cofactor biosynthetic pathway in mouse and human tissues. Using a recently published mouse pancreas atlas (Hrovatin et al, 2023), we observed that *Mocs2* was the highest expressed gene of the pathway across the different cell types of the pancreas (Fig S7A). Expression levels did not vary in different models of diabetes in the same atlas (Fig S7B). The *Suox* enzyme had its highest expression in the endocrine part of the pancreas, albeit at low levels. Compared with the mouse pancreas atlas, we interrogated a recent human scRNA-Seq compilation dataset including 65 pancreata from human donors with/without diabetes from the Human Pancreas Analysis Program (Elgamal et al, 2023). Similar to the mouse pancreas, *MOCS2* had the highest expression across most pancreatic cell types, with the highest

expression being in β-cells and cycling α-cells (Fig S8A–G). On the contrary, *MOCS2* had low basal expression in human liver cell types, whereas *GPHN*, *AOX1*, and *XDH* had the highest expression in hepatocytes in a recently published human liver scRNA-Seq atlas (Fig S8H–N) (Wu et al, 2023 *Preprint*). Overall, we observed a high gene expression of *Mocs2* in the mouse and human endocrine pancreas compared with the rest of the genes of the Moco biosynthetic pathway.

### Phenotypic characterization of *Mocs2* mutant mice

To examine whether any of the molybdenum-related phenotypes observed in the zebrafish model could be translated to mice, we used the deep phenotyping of a *Mocs2* mutant mouse line generated as part of the international mouse phenotyping consortium (Groza et al, 2023). Homozygous Mocs2 mutants were embryonically lethal showcasing the importance of this pathway for proper embryonic development. Therefore, we assessed whether any metabolic phenotypes were present in the heterozygous mice (*Mocs2*$^{+/-}$) between 12 and 16 wk of age. Pancreatic and liver morphology, as evaluated in hematoxylin–eosin-stained sections, was unchanged between WT and *Mocs2*$^{+/-}$ (Fig S9A, B, D, and E). In the endocrine pancreas, immunohistochemical staining of insulin and glucagon demonstrated normal protein patterns and correct mouse islet morphology (Fig S9C and F). We further characterized the terminal differentiation of α-cells and β-cells using the Ucn3 immunostaining that marks mature mouse β-cells (Blum et al, 2012). Regardless of the sex and genotype, Ucn3 was absent from α-cells suggesting correct lineage allocation (Fig 4A–H). Overall, no obvious morphological abnormality was observed in the islets (Figs 4A–H and S9). The relative β-cell area was unchanged (Fig S9G), and we could not identify any proliferating β-cells using Ki67 immunostaining. Then, we assessed metabolic and glucose phenotypes of these mice in both sexes. There were no differences in the body weight between WT and *Mocs2*$^{+/-}$ mice (Fig 4I–K). Overall, *Mocs2*$^{+/-}$ mice showed a trend toward higher fasting glucose, as well as signs of glucose intolerance indicated by slightly delayed glucose clearance in an intraperitoneal glucose tolerance test (IPGTT) (particularly in male mice), but the results did not reach statistical significance (Fig 4M and P). This is due to high variability in the glucose measurements of the *Mocs2*$^{+/-}$ similar to the observed phenotypes of the zebrafish larvae. No phenotypic changes were observed in female *Mocs2*$^{+/-}$ mice, suggesting that there might be sex-specific glucose effects (Fig 4L–Q). Fasting glucose levels and IPGTT approached statistical significance when the results from both sexes were combined (Fig 4N and Q). Lastly, we also examined global markers of liver cell malfunction including alanine aminotransferase (ALAT), bilirubin, and alkaline phosphatase (ALP). No overt changes were observed in ALAT levels (Fig S10A–C). Similar to the glucose phenotype, male mice appeared to have altered biochemical profiles for total bilirubin levels and ALP activity (Fig S10D–I). Increased ALP activity in males could signal liver or bone defects in the *Mocs2*$^{+/-}$ mice (Fig S10H), whereas reduced bilirubin levels are harder to interpret (Fig S10E). A summary of all the clinical parameters assessed for this experiment can be found in Supplemental Data 2. Overall, the deep phenotyping of the *Mocs2*$^{+/-}$ mice showed a trend toward sex-specific (male mice)

altered glucose metabolism and liver alterations, in agreement with the variable phenotype observed in the zebrafish model.

## Discussion

In this work, we aimed to characterize the liver–pancreas crosstalk and how this could affect β-cell regeneration in the highly regenerative zebrafish model. Our results indicate that there is no direct reprogramming of hepatocytes to β-cells in vivo. Previous reports showed that these reprogramming events were based on the overexpression of the fate determinant transcription factor *Pdx1* in mice (Ferber et al, 2000; Ber et al, 2003). It was also reported that the ductal cells of the liver could be reprogrammed to β-cells upon genetic manipulation in mice (Banga et al, 2012). It needs to be noted that the ductal system of the liver and the pancreas is derived from a common progenitor and that the ductal cells of the pancreas are known to contribute to β-cell regeneration in zebrafish (Delous et al, 2012; Maddison & Chen, 2012; Manfroid et al, 2012; Delaspre et al, 2015; Ghaye et al, 2015; Liu et al, 2018; Karampelias et al, 2021, 2022; Mi et al, 2023; Massoz et al, 2024). Whether or not the intrahepatic duct could differentiate into β-cells in zebrafish remains to be seen.

As part of this work, we characterized the transcriptome of isolated hepatocytes from 2-mo-old zebrafish after β-cell ablation. Our data revealed that glucose and lipid metabolism are the most affected biological processes in the absence of β-cells/insulin and associated increased glucose levels. These data confirm the role of insulin as a master regulator of metabolism in the zebrafish liver and stress the conservation of its action to the mammalian orthologue (Saltiel, 2021). However, the number of genes significantly changed was not dramatic with only 51 significantly up-regulated genes discovered in our analysis, suggesting that perhaps more time is needed after β-cell ablation for additional genes to be up-regulated. This could be attributed to low statistical power because of the small number of biological replicates of our study, which is a potential limitation of our work. This observation is strengthened by the fact that both *serpinb1* and *igfbp1a*, genes encoding proteins that were previously implicated in β-cell regeneration and secreted from the liver, were non-significantly up-regulated in our dataset (El Ouaamari et al, 2016; Lu et al, 2016). Only two of the up-regulated, evolutionarily conserved genes had the potential to code for circulating proteins, but none of them stimulated β-cell regeneration when overexpressed.

We also performed a genetic screen by overexpressing the up-regulated enzymes in the hepatocytes. We reasoned that in this way we could affect the level of a circulating metabolite that might stimulate β-cell regeneration and/or glucose levels. Our screen revealed a role for the short isoform of *mocs2* in β-cell regeneration and lowering of glucose. Mocs2 is an enzyme involved in the cascade responsible for the generation of the molybdopterin metabolite and intermediate of the molybdenum cofactor that donates the metal molybdenum to enzymes that need it for their action, including sulfite oxidases and xanthine oxidoreductases. The *mocs2* gene is bicistronic, and two isoforms are transcribed. The long and the short isoforms come together to form a

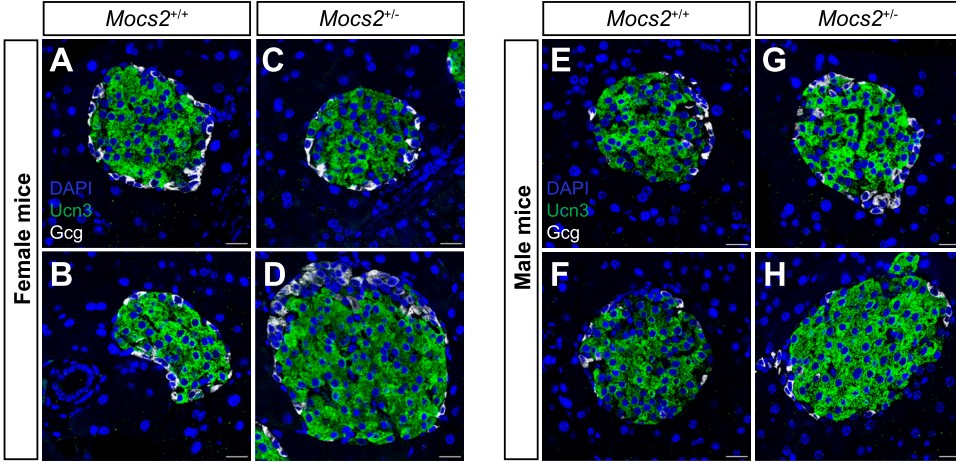

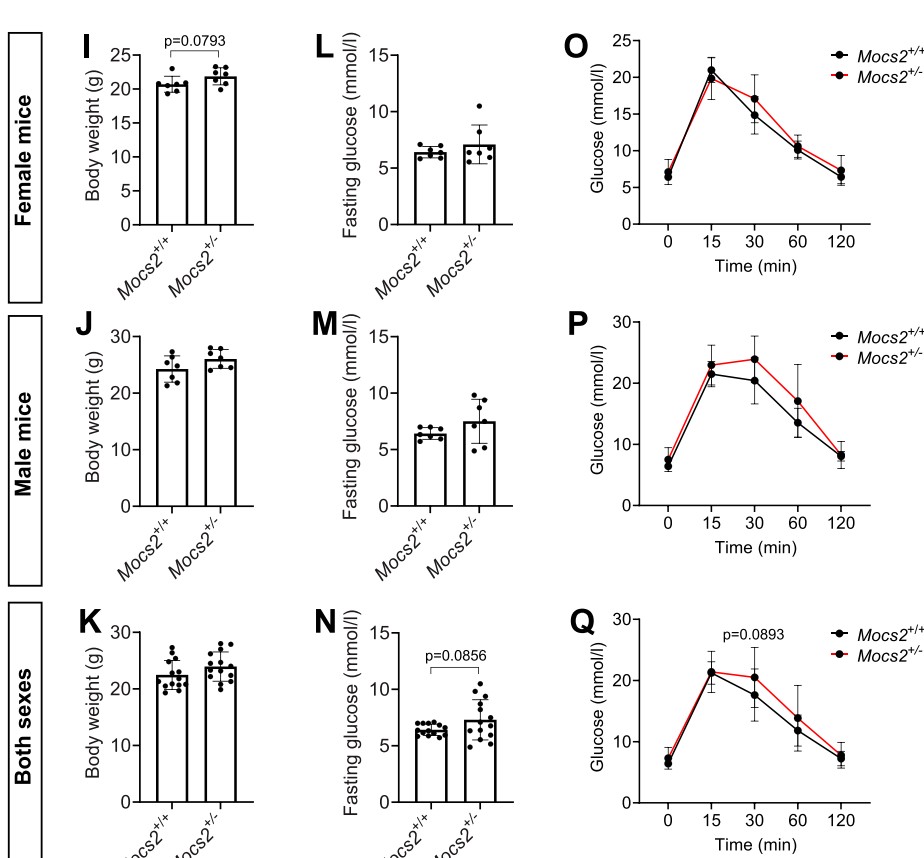

**Figure 4. Phenotyping of *Mocs2*⁺/⁻ mice. (A, B, C, D, E, F, G, H)** Single-plane confocal images of mouse islets from female *Mocs2*⁺/⁺ (A, B) and *Mocs2*⁺/⁻ (C, D), as well as male *Mocs2*⁺/⁺ (E, F) and *Mocs2*⁺/⁻ (G, H), stained for Gcg and Ucn3, and counterstained with DAPI. Scale bar, 20 μm. **(I, J, K)** Body weight measurement before the start of the fasting period for the intraperitoneal glucose tolerance test from female (I), male (J), and combined data (K) for *Mocs2*⁺/⁺ and *Mocs2*⁺/⁻ mice. A Mann–Whitney test was used to assess significance. **(L, M, N)** Fasting glucose measurement from female (L), male (M), and combined data (N) for *Mocs2*⁺/⁺ and *Mocs2*⁺/⁻ mice. A *t* test was used to assess significance. **(O, P, Q)** Intraperitoneal glucose tolerance test results from female (O), male (P), and combined data (Q) for *Mocs2*⁺/⁺ and *Mocs2*⁺/⁻ mice. Two-way ANOVA followed by Šidák's multiple test was used to assess significance.

heterotetrameric complex responsible for the generation of molybdopterin (Mayr et al, 2021). We observed that only the short isoform of the *mocs2* gene, which is the catalytic subunit of the complex, could increase β-cell regeneration. In addition, sodium molybdate treatment recapitulated the increase in β-cell regeneration observed in our genetic model. Interestingly, sodium molybdate treatment has been shown to lower glucose levels in a mouse model of obesity, in a rat model of β-cell ablation using streptozotocin, and in *Drosophila melanogaster* (Ozcelikay et al, 1996; Reul et al, 1997; Perkhulyn et al, 2017). Our results from zebrafish larvae did not recapitulate the glucose-lowering effect of sodium molybdate treatment, which might be due to dosing or stage (because we did see a glucose-lowering effect by *mocs2* overexpression). There is also evidence for a role of this pathway in human metabolism, as in a recent case–control study molybdenum was inversely correlated with the development of metabolic

syndrome (Li et al, 2021). Our data expand on these observations and suggest that perhaps a part of the lower glucose effect in these models might be due to an increased β-cell mass/functionality.

Global *Mocs2* knockout mice were embryonically lethal highlighting its conserved role throughout evolution (Mayr et al, 2021). In this context, our global knockout confirmed the importance of this pathway for mammalian development. In *Mocs2*$^{+/-}$ mice, we observed trends for altered glucose regulation and potential liver phenotypes pointing to potential involvement of this pathway in glucose control also in mammals. Phenotypes have been observed in heterozygote animals previously in the context of the pancreas; for example, heterozygote *Sox9* mice showed glucose intolerance and endocrine formation defects, suggesting that they can provide insights into important biology (Seymour et al, 2008; Puri et al, 2024). However, we acknowledge the need for tissue-specific (liver and β-cell) full knockout of *Mocs2* to dissect its role in each tissue and how this could affect glucose metabolism and β-cell mass. We propose that it could be a future direction, following up on our study with spatiotemporal genetic characterization of the pathway in vivo.

In both the zebrafish and mouse genetic perturbations of Mocs2, the observed phenotypes were variable between biological replicates, suggesting additional factors might be needed to strengthen the β-cell mass and glucose-lowering phenotypes. Most of the enzymes that need the molybdopterin paralogues are involved in the regulation of reactive oxygen species (ROS) biology (Sun et al, 2020). ROS have been implicated as regulators of β-cell mass and function in various experimental models. Recently, it was shown that ROS is involved in β-cell proliferation during early postnatal mouse developmental and under conditions that mimic certain aspects of diabetes (Zeng et al, 2017; Vivoli et al, 2023). Similarly, the levels of ROS in the zebrafish larvae were postulated to be important for regulating β-cell proliferation, such that too low or high levels do not stimulate β-cell proliferation (Alfar et al, 2017). Moreover, recent evidence revealed a new molecular mechanism of redox regulation of insulin secretion (Ferdaoussi et al, 2015; Merrins et al, 2022; Lin et al, 2024). Therefore, we hypothesize that the variability in our genetic and chemical models of Moco biosynthesis could stem from a variable level of generated ROS.

Although our study is aimed to mainly describe the liver–pancreas crosstalk during β-cell regeneration, there are several potential molecular mechanisms (in addition to pancreatic ROS generation) for the glucose/β-cell alterations observed upon Moco pathway perturbations. Our scRNA-Seq data reanalysis showed a high expression of *Mocs2* in mice and human β-cells. There has been a report suggesting that Mocs2 is part of a multiprotein complex that regulates sulfur amino acid metabolism and through that reduces ROS, adding an extra layer of complexity to the pathway (Suganuma et al, 2018). Whether this newly discovered Mocs2 protein complex is active and has a role in glucose regulation remains to be seen. Lastly, recent debated evidence implicates artemether, an antimalarial drug, in β-cell regeneration in mice and endocrine lineage decisions in a human stem cell differentiation model (Li et al, 2017; van der Meulen et al, 2017; Ackermann et al, 2018; Zhang et al, 2023; Canan et al, 2024 *Preprint*). One study proposed that artemether targets gephyrin, a protein that is part of the molybdenum biosynthetic cascade (Li et al, 2017). In our work,

we did not observe any transdifferentiation toward β-cells (unpublished observations) in our transgenic model, arguing for our phenotype being independent of the artemether–gephyrin interaction. Overall, there are several possible molecular mechanisms of the Moco biosynthetic pathway in β-cell biology and glucose regulation that remain to be explored in subsequent studies.

In summary, our discovery study characterizes the liver-to-pancreas axis under conditions of β-cell ablation in the zebrafish. Our genetic screen reveals a previously unexplored role of the molybdenum biosynthetic pathway in β-cell regeneration and glucose regulation. Together with previous findings, our observations encourage further exploration of the role of *mocs2* in glucose control and diabetes.

# Materials and Methods

### Zebrafish transgenic lines

Zebrafish experiments were conducted in compliance with local guidelines and approved by Stockholms djurförsöksetiska nämnd. The previously generated transgenic lines used in this study were as follows: *Tg(ins:flag-NTR)*$^{s950}$ (Andersson et al, 2012), *Tg(ins:CFP-NTR)*$^{s892}$ (Curado et al, 2007), *Tg(ins:Kaede)*$^{s949}$ (Andersson et al, 2012), *Tg(ins:H2BGFP)*$^{KI112}$ (Karampelias et al, 2021), *Tg(fabp10a:Cre)*$^{s955}$ (Ni et al, 2012), *Tg(fabp10a:GFP)*$^{as3}$ (Her et al, 2003), *Tg(fabp10a:CFP-NTR)*$^{s931}$ (Choi et al, 2014), *Tg(ins:venus-geminin)*$^{ki107Tg}$ (Lu et al, 2016), and *Tg(-3.5ubb:loxP-EGFP-loxP-mCherry)*$^{cz1701}$ (Mosimann et al, 2011) referred to as *ubi:Switch*. As part of this work, we generated a stable transgenic line overexpressing the short isoform of *mocs2*, *Tg(fabp10a:smocs2)*$^{KI119}$. Official zebrafish line names were obtained from the ZFIN database (Bradford et al, 2022).

### Mocs2 mouse model generation

The *Mocs2*$^{+/-}$ mouse line (*C57BL/6N-Mocs2*$^{tm1b(EUCOMM)Wtsi}$/*Ieg*) was constructed using the IMPC "knockout first" targeting strategy at Helmholtz Zentrum München, Germany, as follows. Mocs2 mutant mice were generated by allele conversion of a C57BL/6N-*Mocs2*$^{tm1a(EUCOMM)Wtsi}$/*Ieg* mouse line originating from the EUCOMM ES clone EPD0560_5_C09 (for a clone construction overview, see https://www.mousephenotype.org/data/genes/MGI:1336894#order).

Tm1b was produced by crossing *C57BL/6N-Mocs2*$^{tm1a(EUCOMM)Wtsi}$/*Ieg* mice with a ubiquitously active general Cre-deleter mouse line (*C57BL/6NTac-Gt(ROSA)26Sor*$^{tm16(cre)Arte}$) resulting in a deletion of exons 3–5 of Mocs2 (ENSMUSE00001315940, ENSMUSE00001207721, and ENSMUSE00001301379) and the neomycin cassette. The mice were genotyped to verify the mutation (genotyping protocol is available at https://infrafrontier.eu/wp-content/uploads/genotype_protocols/EM09099_geno.pdf).

Heterozygous mice were intercrossed to generate mutant KO mice with WT litter mate controls for experimental analysis. There was evidence of postnatal lethality of homozygous offspring. After birth, 38% (15 of 50) offspring died. Of the remaining offspring, 46% (16 of 35) *Mocs2*$^{+/+}$ and 54% (19 of 35) *Mocs2*$^{+/-}$ were present and viable.

The Mocs2 mouse line is available at the European Mouse Mutant Archive (EMMA/Infrafrontier) (https://www.infrafrontier.eu/emma/strain-search/straindetails/?q=9099).

Mice were housed in IVC cages with water and standard mouse chow available ad libitum according to the European Union directive 2010/63/EU and GMC housing conditions (www.mouseclinic.de). All tests were approved by the responsible authority of the district government of Upper Bavaria.

## Mouse experimental procedures

From the age of 8–16 wk, 14 *Mocs2*$^{+/−}$ mice (7m/7f) and *Mocs2*$^{+/+}$ controls (7m/7f) were phenotyped systematically in the German Mouse Clinic (GMC) as described previously (Fuchs et al, 2011, 2018) and in accordance with the standardized phenotyping pipeline of the IMPC (IMPReSS: https://www.mousephenotype.org/impress/index).

Altered glucose metabolism was examined using the IPGTT at the age of 13 wk. Glucose was administered intraperitoneally (2 g/kg i.p.) after a 16-h withdrawal of food with body weight being determined before and after food withdrawal and glucose levels being measured before and at 15, 30, 60, and 120 min after glucose injection. Blood glucose concentrations were assessed in blood collected from the tail vein with the Accu-Chek Aviva Connect glucose analyzer (Roche/Mannheim).

At the age of 16 wk, the final blood samples were collected from the retrobulbar vein plexus under isoflurane anesthesia in Li-heparin–coated tubes (Li1000A; KABE-Labortechnik). The samples were centrifuged at 5,000$g$ for 10 min at 8°C, and plasma was separated within 2 h after blood collection. Clinical chemistry parameters were measured immediately using an AU480 analyzer (Beckman Coulter) and adapted reagent kits from Beckman Coulter according to the manufacturer's instructions, as described previously (Rathkolb et al, 2013).

Body weight was taken weekly before the different testing procedures throughout the whole phenotyping pipeline procedure. Tissues derived from WT and mutant mice were fixed in neutral buffered formalin, processed, embedded in paraffin, sectioned at 3 $\mu$m, and stained with a standard hematoxylin and eosin protocol. The double immunohistochemical staining was performed using a BOND RX$^m$ (Leica, Germany) automated stainer. Pancreas sections were deparaffinized; antigens were retrieved with citrate buffer for 30 min and blocked for 30 min with a blocking agent. Primary antibodies (insulin rabbit monoclonal, 3,014, 1:3,000; Cell Signaling and glucagon, mouse monoclonal, G2654, 1:1,000; Sigma-Aldrich) were applied and incubated for 60 min followed by secondary antibodies. The detection was performed with DAB and Red BOND polymers. Slides were counterstained with hematoxylin and mounted with a DAKO mounting medium. The slides were imaged with a NanoZoomer S60 scanner. These images were used for quantification of the $\beta$-cell area as the percentage of the total pancreas area using QuPath v0.5.1 (Bankhead et al, 2017). For the immunofluorescence procedure, sections were deparaffinized and rehydrated, and antigens were retrieved with sodium citrate, pH 6, buffer by microwave heating for 15 min. Sections were permeabilized with 0.3% Triton X-100, followed by blocking solution and primary antibody incubation with anti-urocortin 3 (rabbit,

H-019-29, 1:300; Phoenix Pharmaceuticals), anti-glucagon (mouse, sc-514592, 1:400; Santa Cruz), and anti-Ki67 (rabbit, ab15580; Abcam) overnight, followed by a 2-h incubation at room temperature with fluorescence-conjugated secondary antibodies. Imaging was performed using a Leica SP5 confocal microscope.

## Chemical treatments of zebrafish larvae

$\beta$-cell and hepatocyte ablation in the *Tg(ins:CFP-NTR)*, *Tg(ins:flag-NTR)*, or *Tg(fabp10a:CFP-NTR)* was performed by incubating the zebrafish for 24 h with 1 mM (2-mo-old fish) or 10 mM (larvae) MTZ (Sigma-Aldrich) diluted in 1% DMSO (VWR) in facility water (2-mo-old fish) or an E3 solution supplemented with 0.2 mM 1-phenyl-2-thiourea (PTU; Acros Organics) (larvae). Hepatocyte injury was performed by incubating the zebrafish larvae with 10 mM acetaminophen (Sigma-Aldrich) for 24 h. Sodium molybdate (Sigma-Aldrich) was added to the zebrafish water at a final concentration of 10 $\mu$M.

## Immunostaining of zebrafish larvae

Immunofluorescence of zebrafish larvae was described previously (Lu et al, 2016). Primary antibodies were used against GFP (1:500, GFP-1020; Aves Labs), insulin (1:100; custom-made by Cambridge Research Biochemicals), mouse anti-glucagon (1:200, G2654; Sigma-Aldrich), and somatostatin (1:300; DAKO-A0566). Liver area and insulin area were calculated on maximum projections using the Fiji threshold function.

## Gene expression analysis in zebrafish larvae

One larva per tube was used for RNA extraction using Quick-RNA Microprep Kit (R1051; Zymo Research), and cDNA was synthesized using High-Capacity cDNA Reverse Transcription Kit (Applied Biosystems). iTaq Universal SYBR Green Supermix (Bio-Rad) was used for the qPCR, amplification was measured using ViiA 7 Real-Time PCR System (Applied Biosystems), and results were analyzed using the DDCT method (Livak & Schmittgen, 2001). Primer sequences are provided in Table S1.

## Glucose measurements in zebrafish larvae

Glucose levels were measured using the Glucose Colorimetric/Fluorometric Assay kit (BioVision) in pools of four larvae for each time point and condition. Blood glucose measurements in 2-mo-old zebrafish were performed using a standard glucometer (FreeStyle; Abbott). Fish were fasted for 16 h, anesthetized in tricaine (Sigma-Aldrich), and decapitated for blood glucose measurements.

## Hepatocyte isolation and RNA-Seq

Three independent hepatocyte isolations from 2-mo-old zebrafish were performed from three different crosses. $\beta$-Cells were ablated with MTZ for 24 h followed by a washout of the drug, feeding, and regeneration for an additional day. Zebrafish treated with MTZ but without the *Tg(ins:flag-NTR)* transgene were used as controls to

adjust for the effect of the MTZ on the transcriptome. 10–15 zebrafish of mixed sex were euthanized by head decapitation, and the livers were dissected with the help of a fluorescence microscope, as the fish carried the *Tg(fabp10a:GFP)* transgene. Then, livers were cut into smaller pieces, passed through a needle to further disrupt the tissue, and incubated with TrypLE for 45 min at 37°C until a single suspension was created. TrypLE was inactivated using fetal bovine serum, and the single suspension was sorted using the FACSAria III instrument and the FITC-A filter to sort the GFP⁺ cells.

RNA was extracted from the sorted cells using RNAqueous-Micro Total Isolation Kit (Invitrogen). The quality of the RNA was assessed, and the first control sample was discarded as the RIN value was too low. Libraries were prepared using the TruSeq Stranded mRNA kit (Illumina) and sequenced on an Illumina HiSeq 2500 instrument. Reads were mapped using TopHat 2.0.4 (Trapnell et al, 2012) to the GRCz10 zebrafish genome assembly. Differential expression analysis was conducted using the DESeq package on the R environment (Anders & Huber, 2010). Gene ontology analysis was performed using the Panther online database (version 15.0) (Thomas et al, 2003; Mi et al, 2010). To identify potentially secreted proteins coded from the up-regulated genes upon hepatocyte ablation, we used the SignalP (4.0) bioinformatics tool (Petersen et al, 2011) to assess the presence of a signal peptide targeting the protein for secretion. After the first screen, we excluded the transmembrane, extracellular matrix proteins, and proteins that do not have orthologues in humans. In addition, we assessed the up-regulated genes for targeted secretion through unconventional secretory pathways using the OutCyte 2.0 bioinformatics tool (Zhao et al, 2019). Only one protein coded by the gene *ENSDARG00000086518* was predicted to be unconventionally secreted, but it was not conserved to mammals, and therefore, we did not pursue it for further analysis.

### Cloning of selected genes and genetic screen

Cloning of *sdf2l1* and the selected enzymes was performed using the Gateway system. Of note, we did not manage to clone two significantly up-regulated enzymes, namely, *pdzrn3a* and *tor1l1* because of a lack of cDNA amplification. Pooled cDNA from different developmental zebrafish larval stages was used as a template to amplify the gene sequences, which were subsequently cloned into the pDONR221 middle entry vector. Primer sequences for the amplification of the genes are shown in Table S2. We also generated a 5′ entry vector containing the *fabp10a* promoter. Then, we performed three-way recombination reactions using the 5″ entry clone containing the *fabp10a* promoter, the middle entry vectors containing the different genes, the 3′ *polyA* entry vector, and the pDESTtol2CG2 destination vector that carries the *cmcl2:GFP* transgene for selection. The final vectors were sequenced, and then, 15 pg of the vectors together with 20 pg of transposase mRNA was injected into the one-cell-stage zebrafish embryos to induce mosaic overexpression. At 3 dpf, the larvae were screened for the transgene integration by assessing the presence of the *cmcl2:GFP* expression. The regeneration assay was performed by ablating the β-cells between 3 and 4 dpf and manually counting the number of β-cells using a fluorescence microscope at 6 dpf.

### Expression analysis using publicly available scRNA-Seq datasets

For the mouse pancreas gene expression analysis, we used the processed and annotated mouse pancreas scRNA-Seq atlas dataset (Hrovatin et al, 2023), which was downloaded from GEO with the accession number: GSE211799, file name: GSE211799_adata_atlas.h5ad.gz. The .h5ad file was imported into a Jupyter notebook, and violin plots were generated using the Scanpy toolkit (1.9.5) and the "cell_type_integrated_v2_parsed" clustering option (Wolf et al, 2018).

For the human liver gene expression analysis, we used the processed and annotated human liver pancreas scRNA-Seq atlas.h5ad file, downloaded from the preprint associated website (https://liver.unifiedcellatlas.org), file name: "normal_final_webV2.h5ad" (Wu et al, 2023 *Preprint*). We visualized the expression using violin plots with the "level 1" obs clustering option of the file in the Scanpy toolkit (1.9.5).

For the human islet dataset, we used the processed and annotated human islet scRNA-Seq dataset from Elgamal et al (2023), which was downloaded from https://www.gaultonlab.org/pages/Islet_expression_HPAP.html. The file was imported in RStudio using R version 4.2.3 (R Core Team, 2019), and the violin plots were generated with the group.by = "Cell Type" annotation using the Seurat package 4.4.0 (Hao et al, 2021).

### Statistical analysis

Statistical analysis throughout the study was carried out using GraphPad Prism software. *P*-values ≤ 0.05 were considered significant. Normal distribution of data was calculated using a combination of the Shapiro–Wilk and D'Agostino–Pearson tests. *P*-values are reported in the respective figure legends of the experiments where appropriate.

# Data Availability

The data that support the findings of this study are available from the corresponding authors upon reasonable request. The raw reads of the RNA-Seq data are available in the Sequence Read Archive under the accession number PRJNA1103351.

# Supplementary Information

# Acknowledgements

Research in the laboratory of O Andersson was supported by funding from the Swedish Research Council, the Novo Nordisk Foundation, the Diabetes Wellness, the Cancerfonden, and the Strategic Research Programmes in Diabetes at the Karolinska Institutet. E Bachar-Wikström was funded by a postdoctoral fellowship from Novo Nordisk A/S. The German Mouse Clinic was supported by the German Federal Ministry of Education and Research

(Infrafrontier grant 01KX1012 to M Hrabě de Angelis) and the German Center for Diabetes Research (DZD) (to M Hrabě de Angelis). The authors acknowledge support from the Science for Life Laboratory, the Knut and Alice Wallenberg Foundation, the National Genomics Infrastructure funded by the Swedish Research Council, and the Uppsala Multidisciplinary Center for Advanced Computational Science for assistance with massively parallel sequencing (alternatively, genotyping) and access to the UPPMAX computational infrastructure.

## Author Contributions

C Karampelias: conceptualization, data curation, formal analysis, validation, investigation, visualization, methodology, project administration, and writing—original draft, review, and editing.
B Bǎloiu: data curation, formal analysis, validation, and writing—review and editing.
B Rathkolb: data curation, formal analysis, investigation, methodology, and writing—review and editing.
P da Silva-Buttkus: data curation, formal analysis, investigation, methodology, and writing—review and editing.
E Bachar-Wikström: data curation, formal analysis, investigation, and writing—review and editing.
S Marschall: resources, methodology, and writing—review and editing.
H Fuchs: formal analysis, supervision, project administration, and writing—review and editing.
V Gailus-Durner: data curation, formal analysis, supervision, project administration, and writing—review and editing.
L Chu: formal analysis, investigation, and methodology.
M Hrabě de Angelis: resources, supervision, funding acquisition, validation, and project administration.
O Andersson: conceptualization, data curation, formal analysis, supervision, funding acquisition, investigation, visualization, methodology, project administration, and writing—original draft, review, and editing.

## Conflict of Interest Statement

The authors declare that they have no conflict of interest.

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
