## [Reviewer comments · Life Science Alliance]

Life Science Alliance

Examining the liver-pancreas crosstalk reveals a role for molybdenum cofactor in b-cell regeneration

Christos Karampelias, Bianca Băloiu, Birgit Rathkolb, Patricia da Silva-Buttkus, Ety Bachar-Wikström, Susan Marschall, Helmut Fuchs, Valerie Gailus-Durner, Lianhe Chu, Martin Hrabě de Angelis, and Olov Andersson

DOI: <https://doi.org/10.26508/lsa.202402771>

Corresponding author(s): Christos Karampelias, Helmholtz Zentrum München and Olov Andersson, Karolinska Institutet

Review Timeline:

Submission Date:	2024-04-16
Editorial Decision:	2024-06-07
Revision Received:	2024-08-06
Editorial Decision:	2024-08-06
Revision Received:	2024-08-07
Accepted:	2024-08-08

Transaction Report:

June 7, 2024

Re: Life Science Alliance manuscript #LSA-2024-02771

Dr. Christos Karampelias
Helmholtz Zentrum München
Ingolstädter Landstraße 1
Neuherberg 85764
Germany

Dear Dr. Karampelias,

Thank you for submitting your manuscript entitled "Examining the liver-pancreas crosstalk reveals a role for molybdenum cofactor in b-cell regeneration" to Life Science Alliance. The manuscript was assessed by expert reviewers, whose comments are appended to this letter. We invite you to submit a revised manuscript addressing the Reviewer comments.

Thank you for this interesting contribution to Life Science Alliance. We are looking forward to receiving your revised manuscript.

Sincerely,

B. MANUSCRIPT ORGANIZATION AND FORMATTING:

Reviewer #1 (Comments to the Authors (Required)):

This paper used an in vivo model of beta cell regeneration to investigate inter-organ crosstalk and the potential for hepatocytes to contribute to regeneration directly by transdifferentiation or indirectly through the secretion of protein factors or metabolites. The authors convincingly show that hepatocyte transdifferentiation does not contribute to beta cell regeneration in unperturbed zebrafish using genetic lineage tracing. They further show that liver damage does not compromise beta cell development. Gene expression analysis of isolated hepatocytes from young adult zebrafish +/- beta cell ablation identified differentially expressed genes. Focusing on the upregulated genes set predicted secreted proteins and enzymes were functionally tested in the larval beta cell regeneration model and the molybdenum synthesis pathway was implicated. Commendably, the authors acknowledge that mocs2 overexpression or sodium molybdate supplementation had variable effects on beta cell regeneration across experimental batches with similar variability seen in a mocs2 heterozygous mouse model.

Outstanding questions that should be addressed:

- 1) Does liver damage affect beta cell regeneration? It seems that acetaminophen treatment could be administered during beta cell regeneration in the MTZ/ins:NTR model.
- 2) How were the secreted liver proteins predicted? Many cytoplasmic proteins that lack canonical signal sequences can be released in exosomes.
- 3) Further insight into the variability of the mocs2 OE/supplementation phenotype. While identifying contributors to batch variability can be challenging, the authors could provide all of the data from non-significant runs in the supplemental data so the reader can appreciate the number of trials needed to produce the significant results in the main figures. Perhaps loss-of-function experiments would be less susceptible to these unidentified variables? The mocs2 crispant phenotype should be tested in the beta cell regeneration assay.

Minor points:

- 1) line 150 - ablation system instead of "model"
- 2) line 207-208 and figures. Please check the nomenclature, small and big isoform is uncommon usage versus short and long isoforms.
- 3) line 275 - perhaps "interrogated" instead of "enquired"

Referee Cross-Comments

I agree that the manuscript can be improved by addressing some of Reviewer #2' comments. However some points are out of scope or not required for an in vivo study of this type. Specifically:

#1 The authors have shown here and in many previous publications that recovery of damaged (de differentiated) beta cells does not make a detectable contribution to proliferation in regenerating islets.

#2 While reductionist systems have value for investigating biology, it is not trivial to recapitulate in vivo regenerative processes in a transwell assay. This is certainly an option for future studies but should not be required to publish results from intact in vivo models.

Reviewer #2 (Comments to the Authors (Required)):

Overall it is a very nice study that identified a new factor/signaling pathway in hepatic cells to regulate pancreatic beta cell proliferation. Some concerns need to be addressed:

1. Conclusion on Beta-Cell Proliferation:

The conclusion about the proliferation of beta-cells using EdU is not sufficient as it could not exclude the possibility of injured cells' absorption of EdU during repair. A staining for an endogenous proliferation marker such as Ki-67 is needed to validate the results.

2. Lack of In Vitro Co-Culture Experiment:

The study lacks an in vitro transwell co-culture experiment of hepatic cells and beta cells with proper interference, which is crucial to support the in vivo data and provide important molecular insight. For example, the study identifies a role for the Moco biosynthetic pathway in beta-cell regeneration but there are lack of gain-of-function or loss-of-function experiments done in vitro to assess it functionally.

3. Global Knockout Model Limitation:

The use of a global knockout of MOCS2 is not ideal. The paper should discuss the limitations of this model in more detail, especially if a cell-specific ablation mouse model is not available.

4. Incomplete Immunostaining in Figure 3:

Figure 3's immunostaining lacks nuclear staining with DAPI or Hoechst. Including this would provide a clearer understanding of percentage of the positive cells.

5. Beta-Cell Mass Measurement in Mouse Study:

The mouse study does not measure beta-cell mass, which is a critical parameter for evaluating beta-cell regeneration and function.

6. Inconsistent Phenotypes in Mocs2 Overexpression:

The observed phenotypes in both zebrafish and mouse models of Mocs2 overexpression are variable between biological replicates, which requires further discussion.

7. Lack of Detailed Mechanistic Insight:

While the study identifies genes and pathways involved in beta-cell regeneration, it lacks detailed mechanistic insight into how these genes and pathways interact to influence beta-cell proliferation and glucose metabolism. Deep discussion is needed.

Reviewer #1 (Comments to the Authors (Required)):

This paper used an in vivo model of beta cell regeneration to investigate inter-organ crosstalk and the potential for hepatocytes to contribute to regeneration directly by transdifferentiation or indirectly through the secretion of protein factors or metabolites. The authors convincingly show that hepatocyte transdifferentiation does not contribute to beta cell regeneration in unperturbed zebrafish using genetic lineage tracing. They further show that liver damage does not compromise beta cell development. Gene expression analysis of isolated hepatocytes from young adult zebrafish +/- beta cell ablation identified differentially expressed genes. Focusing on the upregulated genes set predicted secreted proteins and enzymes were functionally tested in the larval beta cell regeneration model and the molybdenum synthesis pathway was implicated. Commendably, the authors acknowledge that mocs2 overexpression or sodium molybdate supplementation had variable effects on beta cell regeneration across experimental batches with similar variability seen in a mocs2 heterozygous mouse model.

We thank the reviewer for the support and constructive feedback on our manuscript.

Outstanding questions that should be addressed:

1) *Does liver damage affect beta cell regeneration? It seems that acetaminophen treatment could be administered during beta cell regeneration in the MTZ/ins:NTR model.*

This is indeed a good point that we have already tried to address. Unfortunately, we tried both acetaminophen and double ablation NTR-MTZ model (NTR expressed in both hepatocytes and beta-cells), but both approaches were too toxic for the zebrafish which died shortly afterwards. Therefore, this experiment is not feasible for viability reasons, suggesting even in the highly regenerating zebrafish system a simultaneous ablation of hepatocytes and beta-cells is not viable. We have updated the text to convey this information (line 153 in revised manuscript).

2) *How were the secreted liver proteins predicted? Many cytoplasmic proteins that lack canonical signal sequences can be released in exosomes.*

For identifying secreted liver proteins, we used SignalP bioinformatics approach to check for signal peptides targeting proteins for secretion. From this first list we excluded transmembrane, extracellular matrix proteins, and proteins that are not conserved to mammals.

Following this comment, we have now checked for exosome-dependent secretion using the Outcyte tool and identified 1 proteins with gene name:

ENSDARG00000086518

as a potential exosome secreted proteins, however it is not conserved to mammals so we did not pursue it further. We have updated the methods part of our manuscript (line 605) to accurately report the secreted protein identification process.

3) *Further insight into the variability of the mocs2 OE/supplementation phenotype. While identifying contributors to batch variability can be challenging, the authors could provide all of the data from non-significant runs in the supplemental data so the reader can appreciate the number of trials needed to produce the significant*

results in the main figures. Perhaps loss-of-function experiments would be less susceptible to these unidentified variables? The *mocs2* crispant phenotype should be tested in the beta cell regeneration assay.

*We appreciate the comment and have now added all the individual experiments from figure 3 as a supplementary file to clarify the variability (New supplementary Figure 3). Furthermore, we have indeed previously tried to generate CRISPR knockout of the short subunit of *mocs2*. The challenge we faced was to identify good targeting sequences in the relatively small nucleotide sequence of the short isoform of the *mocs2* gene (261 nucleotides length in 3 exons), i.e. to keep the long isoform intact. Nevertheless, we designed 2 gRNAs and saw that at least one of them caused mutations in heterozygote Founder zebrafish. Unfortunately, we had trouble generating a stable line, perhaps due to a potential embryonic lethal phenotype; similar to the mouse model in which the mutant *mocs2* gene is shown to cause severe developmental disorders. Further, since only one of the gRNAs produced one documented mutation, it would be hard to generate crispants with high mosaicism, and thus interpret potential results. Moreover, we have actually never worked with mutants (or small molecules) that consistently reduce beta-cell regeneration (which is the tentative result of mutating the short isoform of *mocs2*), as data might be more reliable if having a positive effect.*

Minor points:

- 1) line 150 - ablation system instead of "model"
- 2) line 207-208 and figures. Please check the nomenclature, small and big isoform is uncommon usage versus short and long isoforms.
- 3) line 275 - perhaps "interrogated" instead of "enquired"

All points are corrected in the text and figures.

Referee Cross-Comments

I agree that the manuscript can be improved by addressing some of Reviewer #2' comments. However some points are out of scope or not required for an in vivo study of this type. Specifically:

#1 The authors have shown here and in many previous publications that recovery of damaged (de differentiated) beta cells does not make a detectable contribution to proliferation in regenerating islets.

#2 While reductionist systems have value for investigating biology, it is not trivial to recapitulate in vivo regenerative processes in a transwell assay. This is certainly an option for future studies but should not be required to publish results from intact in vivo models.

Reviewer #2 (Comments to the Authors (Required)):

Overall it is a very nice study that identified a new factor/signaling pathway in hepatic cells to regulate pancreatic beta cell proliferation. Some concerns need to be addressed:

We appreciate the supportive and thorough review of our work.

1. Conclusion on Beta-Cell Proliferation:

The conclusion about the proliferation of beta-cells using EdU is not sufficient as it could not exclude the possibility of injured cells' absorption of EdU during repair. A staining for an endogenous proliferation marker such as Ki-67 is needed to validate the results.

We have now added data using a transgenic fishline indicating beta-cell proliferation (ins:venus-geminin) following beta cell ablation and sodium molybdate treatment, showing similar trends as the EdU results. Ki67 staining was not performed as it is not working in zebrafish. Further we show that proliferation is not affected at the basal state during development of zebrafish, where cellular damage and potential DNA repair is minimal. We further note that beta-cell death does not affect the proliferation and EdU incorporation in neighboring cells (alpha-delta cells), as we also have shown in our previous work (new supplementary figure 4). Lastly, the EdU staining occupied the whole nucleus while if it was incorporation due to cell death we would expect speckles of EdU in the apoptotic nuclei.

2. Lack of In Vitro Co-Culture Experiment:

The study lacks an in vitro transwell co-culture experiment of hepatic cells and beta cells with proper interference, which is crucial to support the in vivo data and provide important molecular insight. For example, the study identifies a role for the Moco biosynthetic pathway in beta-cell regeneration but there are lack of gain-of-function or loss-of-function experiments done in vitro to assess it functionally.

We agree with the reviewer that these are indeed interesting experiments. However given the scope of the current work, we reason that would be a topic of a follow up work where we will have the necessary time to devote into a thorough characterisation of both transwell systems as well as conditioned media from hepatocyte lines (both mouse and human) and treatment of isolated beta cells. Additionally, one might face an extra level of complexity in these types of validation experiments, e.g. hypothesize that the molybdenum cofactor formed the liver could signal to circulating immune cells that in turn induce beta-cell regeneration, requiring an even more complex validation system not be possible in vitro.

3. Global Knockout Model Limitation:

The use of a global knockout of MOCS2 is not ideal. The paper should discuss the limitations of this model in more detail, especially if a cell-specific ablation mouse model is not available.

We agree with the reviewer that the global knockout of the MOCS2 knockout mice is not ideal, mainly for two reasons: 1) given the lethality of the homozygote mutant; 2) a tissue specific knockout model could perhaps better dissect the phenotypes and

strengthen the liver-pancreas axis. We have added a paragraph to the discussion and highlight this issue and acknowledge the shortcomings of the heterozygous mice. We will consider detailed breeding schemes (to generate floxed mice that can be crossed to different Cre-drivers) in a follow up work on the topic.

4. Incomplete Immunostaining in Figure 3:

Figure 3's immunostaining lacks nuclear staining with DAPI or Hoechst. Including this would provide a clearer understanding of percentage of the positive cells.

*Unfortunately, we had difficulty costaining with DAPI/Hoechst in the EdU assay protocol and therefore we do not currently have such pictures to update the figure with. However, it is not necessary to have a general nuclear stain to calculate the percentage of proliferating beta-cells as we mark the beta-cells with nuclear fluorescence, using the *ins:H2B-GFP* line. To address this comment, we have updated the text to report the percentage of beta-cell proliferation and have added the quantifications of each independent experiment from Figure 3 to a new supplementary Figure 3 reporting the percentages as well. Additionally, we show a *TO-PRO-3* counterstaining in the new *ins:venus-geminin* experiment (Supplementary Figure 4) that shows the overall morphology of the pancreas is not affected by sodium molybdate treatment. We hope that these revisions have clarified the issue.*

5. Beta-Cell Mass Measurement in Mouse Study:

The mouse study does not measure beta-cell mass, which is a critical parameter for evaluating beta-cell regeneration and function.

*We now provide the % of beta-cell area (compared to total pancreas area) in Supplementary Figure G for the immunostained pancreata. While we understand this is not equivalent to total beta-cell mass measurements, the pancreas weights were not collected at the time of the phenotyping in the international consortium, live mice are not currently available, and the rederivation process is lengthy. We also assessed beta-cell proliferation with Ki67 staining in these *WT* and *Mocs2 +/-* mice, but could not find any proliferating beta cell in any of the 8 examined mice.*

6. Inconsistent Phenotypes in *Mocs2* Overexpression:

The observed phenotypes in both zebrafish and mouse models of *Mocs2* overexpression are variable between biological replicates, which requires further discussion.

We further expand on possible explanations in the discussion and now provide all the individual experiments in supplementary figure 3 for the reader to fully understand the variability of the phenotype.

7. Lack of Detailed Mechanistic Insight:

While the study identifies genes and pathways involved in beta-cell regeneration, it lacks detailed mechanistic insight into how these genes and pathways interact to influence beta-cell proliferation and glucose metabolism. Deep discussion is needed.

We agree with the reviewer that our work is mainly an observational study aimed to perform a first broad screening characterization of the liver-to-pancreas crosstalk

using a highly regenerative model. In an attempt to probe the mechanism of the Moco pathway in beta-cell regeneration, in the revised manuscript, we performed and included qPCR analysis for the main genes involved in the Moco biosynthetic pathway, together with genes that utilize Moco for their function and key glycolytic/gluconeogenic genes in the zebrafish larvae during the regeneration period (new Supplementary Figure 5). We did not identify any major differences in the Moco pathway, but we observed a downregulation of certain Moco-utilizing enzymes in the genetic but not in the chemical treatment model, something that could potentially explain the differences in the glucose lowering phenotypes in the zebrafish. Together with our initial RNA-Seq, and the variability between experimental replicates we agree that there are more mechanistic details needed to understand the crosstalk on the tissue level (how the crosstalk takes place) and on the molecular level. Further, our single cell RNA-Seq analysis of published datasets suggested a high expression of MOCS2 specifically in beta-cells. For this reason, we have now supplemented our discussion with recently published work suggesting an independent role of Mocs2 outside of the Moco biosynthetic pathway that warrants further investigation. Moreover, we have added sentences to the discussion to speculate more about potential mechanisms (short lived ROS in circulation, potential crosstalk with artemether and gephyrin related beta-cell regeneration) of our work. Lastly, we attach a preliminary experiment that further showcases the variability of ROS generation following sodium molybdate treatment in HEK293 cells, showing that at least some part of ROS production is affected by it. We hope that the revised discussion provides a relevant context to questions that might arise from our initial discovery work, and properly place our study in the literature.

Figure: CellRox (a ROS sensor) incorporation in HEK293 cells following 24 hours treatment with 10 μ M of sodium molybdate followed by FACS analysis. $n=3$

August 6, 2024

RE: Life Science Alliance Manuscript #LSA-2024-02771R

Dr. Christos Karampelias
Helmholtz Zentrum München
Ingolstädter Landstraße 1
Neuherberg 85764
Germany

Dear Dr. Karampelias,

Thank you for submitting your revised manuscript entitled "Examining the liver-pancreas crosstalk reveals a role for molybdenum cofactor in b-cell regeneration". We would be happy to publish your paper in Life Science Alliance pending final revisions necessary to meet our formatting guidelines.

- please be sure that the authorship listing and order is correct
- please upload your main manuscript text as an editable doc file
- please upload all figure files as individual ones, including the supplementary figure files; all figure legends should only appear in the main manuscript file after the references section
- please add the Twitter handle of your host institute/organization as well as your own or/and one of the authors in our system
- please upload your Tables in editable .doc or Excel format. They can be included at the bottom of the main manuscript file or sent as separate files.
- please add callouts for Figures S1A-C; S3A-F; S5A-H and K-X; S9A-B and D-E; S10A-D and F, G, I to your main manuscript text

A. FINAL FILES:

B. MANUSCRIPT ORGANIZATION AND FORMATTING:

Sincerely,

August 8, 2024

RE: Life Science Alliance Manuscript #LSA-2024-02771RR

Dr. Christos Karampelias
Helmholtz Zentrum München
Ingolstädter Landstraße 1
Neuherberg 85764
Germany

Dear Dr. Karampelias,

Thank you for submitting your Research Article entitled "Examining the liver-pancreas crosstalk reveals a role for molybdenum cofactor in b-cell regeneration". It is a pleasure to let you know that your manuscript is now accepted for publication in Life Science Alliance. Congratulations on this interesting work.

DISTRIBUTION OF MATERIALS:

Again, congratulations on a very nice paper. I hope you found the review process to be constructive and are pleased with how the manuscript was handled editorially. We look forward to future exciting submissions from your lab.

Sincerely,
